# The accuracy of nucleic acid amplification tests (NAATs) in detecting human intestinal nematode infections: A protocol for a systematic review and meta-analysis

**Nalini Kaushalya Jayakody**[1,2]*, **Poornima Lakshini Kumbukgahadeniya**[2], **Anjana Silva**[2], **Nuwan Darshana Wickramasinghe**[3], **Susiji Wickramasinghe**[4], **Donald Peter McManus**[5], **Kosala Gayan Weerakoon**[2,5]*

**1** Faulty of Medicine, Department of Parasitology, Wayamba University of Sri Lanka, Kuliyapitiya, Sri Lanka, **2** Faculty of Medicine and Allied Sciences, Department of Parasitology, Rajarata University of Sri Lanka, Saliyapura, Sri Lanka, **3** Faculty of Medicine and Allied Sciences, Department of Community Medicine, Rajarata University of Sri Lanka, Saliyapura, Sri Lanka, **4** Faculty of Medicine, Department of Parasitology, University of Peradeniya, Kandy, Sri Lanka, **5** QIMR Berghofer Medical Research Institute, Brisbane, Australia

* kosalagadw83@gmail.com (KGW); jayakodynalini@wyb.ac.lk (NKJ)

**Data Availability Statement:** This protocol was created in April 2022. The review team will then

## Abstract

Human intestinal nematode infections are a global public health issue as they can result in considerable morbidity in infected individuals, mainly in developing countries. These infections continue to go undiagnosed, as they tend to be mainly endemic in resource-poor communities where there is a shortage of experienced laboratory staff and relevant diagnostic technologies. This is further exacerbated by the nature of intermittent shedding of eggs and larvae by these parasites. Diagnostic methods range from simple morphological identification to more specialised high-throughput sequencing technologies. Microscopy-based methods, although simple, are labour-intensive and considerably less sensitive than molecular methods which are rapid and have high levels of accuracy. Molecular methods use nucleic acid amplification (NAA) to amplify the deoxyribonucleic acid (DNA) or ribonucleic acid (RNA) fragments of the parasite to detect and determine its presence using different technologies (NAAT). They have increased the sensitivity of detection and quantitation of intestinal nematode infections, especially in low infection intensity settings. The absence of a gold standard test limits current diagnosis and, in turn, restricts intervention measures and effective control efforts. The objective of this review is to determine the accuracy of NAATs in detecting human intestinal nematode infections using Kato-Katz as the reference test for the most common soil-transmitted helminth (STH) infections and the scotch tape test for enterobiasis and Baermann method for strongyloidiasis. Relevant studies will be identified by searches in electronic databases. Two reviewers will independently screen the literature against eligibility criteria. The methodological quality of studies will then be appraised by two reviewers using the Quality Assessment of Diagnostic Accuracy Studies (QUADAS-2) tool. Discrepancies will be addressed by a third reviewer. The true positives, false positives, true negatives and false negatives of all the studies will be extracted into contingency tables. In

begin implementing the protocol outlined in this document. No datasets were generated or analysed during the current study. Upon completion of the systematic review, all relevant data from this review protocol will be made public.

**Funding:** This review is a part of an intestinal parasitoses community evaluation project funded by the National Research Council of Sri Lanka (NRC 20-118). However, this particular publication (review protocol) is not funded through this grant. The funder had and will not have a role in study design, data collection and analysis, decision to publish, or preparation of the manuscript.

**Competing interests:** The authors have declared that no competing interests exist.

paired forest plots, study-specific sensitivity and specificity with a 95 per cent confidence interval will be displayed. The systematic review of this protocol will report the diagnostic accuracy of currently available NAATs for the detection of human intestinal nematode infections. This will help healthcare providers and administrators determine the diagnostic method to be used in different clinical and preventive settings.

**Trial registration**: PROSPERO registration number for this protocol is CRD42022315730.

## Introduction

Six intestinal nematodes that commonly infect humans are *Ascaris lumbricoides*, *Trichuris trichiura*, *Ancylostoma duodenale*, *Necator americanus*, *Strongyloides stercoralis* and *Enterobius vermicularis*. Of these, *A. lumbricoides*, *S. stercoralis*, *Ancylostoma* spp., *N. americanus*, and *T. trichiura* are STHs [1, 2]. STH infections are the most common of the neglected tropical diseases (NTDs) globally. They cause an insidious impact on the deterioration of health and quality of life in impoverished populations, which leads to the perpetuation of poverty cycles [3]. Morbidity from STH infections becomes noticeable in a community only when a large number of people have moderate or high worm burdens, which usually happens when the prevalence is high. There has been an increased interest in controlling STH infections since the release of the world health organisation (WHO) road map to fight NTDs in 2012 [4]. The 2021 road map targets to reduce 75% of the Disability-Adjusted Life Years (DALYs) due to NTDs and to eradicate at least two NTDs from the world [5, 6].

Epidemiologically, STH infections are the most widespread among people residing in areas with insufficient sanitary facilities, low socioeconomic conditions and tropical climates [7, 8]. Even though frequent in low and middle-income countries, STH infections are also seen among vulnerable populations in high-income countries [9]. The presence of a wide range of symptoms and clinical signs, the challenges in nematode identification, and the scarcity of up-to-date epidemiological data, have made it difficult to arrive at precise estimates of the global burden due to intestinal nematodes. Insufficient disease monitoring systems have further exacerbated this issue. Consequently, estimating the disease burden depends on approximate values based on the best available data [10]. Current global figures show that around 1.5 billion people are diagnosed as suffering from STH infections, resulting in a disease burden of 5.2 million DALYs [11]. Over 267 million preschoolers and 568 million schoolchildren live in communities where these parasites are heavily transmitted and require treatment and preventive measures [12, 13]. *S. stercoralis* alone affects around 600 million individuals worldwide [14]. It is estimated that STH species infect around 361 million South Asians in 2019 [15]. Intestinal nematode infection burdens are higher in Asia due to the wet and hot climate, lack of access to clean drinking water, poor sanitation and sub-standard hygienic practices [16, 17]. In a national survey undertaken in 2019, the island-wide prevalence of STH infection was 0.97% (95% CI 0·63–1·48) among Sri Lankan schoolchildren aged 5–7 years. The prevalence of high-risk populations was greater than the national prevalence, which was 2.73% (0.75–6.87) in urban slum communities and 9.02% (4.29–18.0) in plantation sector communities [18]. However, these assessments are probably an underestimate of the true picture because they were calculated using microscopy-based methods, which have limited sensitivity in diagnosing intestinal nematode infections [19]. Even though microscopic methods have recognised low sensitivity, they are used in many instances as they are relatively simple and quick to perform,

inexpensive and facilitate direct visualisation of nematode eggs and larval morphology. Individuals with high-intensity infections have higher morbidity, such as stunted physical growth and impaired cognition [17], whereas subjects with low-intensity infections usually do not have any symptoms. Data from the past ten years show a reduction in the STH infection prevalence in upper-middle-income countries, shifting the disease burden to lower-middle-income and low-income countries [20]. DALYs due to STH infections understate the accurate burden of disease, mainly to its incorrect attribution of hookworm-induced anaemia [1, 20].

Diagnostic methods for intestinal nematodes range from simple morphological identification to more specialised technologies, which include microscopic, serological and molecular-based diagnostics. Microscopic diagnostics include methods such as direct wet mount, Kato-Katz, formol ether concentration, spontaneous tube concentration, agar plate culture, water emergence method for detecting *Strongyloides* larvae in stools, Baermann technique, Harada-Mori technique, merthiolate iodine formaldehyde concentration, flotation procedures, McMaster's method, FLOTAC and Mini-FLOTAC techniques, Stoll's dilution egg counting technique.

NAATs are different molecular methods designed to amplify specific DNA or RNA fragments into a higher number of copies in order to determine and characterize their presence [21]. Types of NAATs used for the diagnosis of intestinal nematode infections are polymerase chain reaction (PCR), including conventional single, nested, real-time, multiplex PCR and digital PCR, isothermal assays like loop-mediated isothermal amplification (LAMP) quantitative nucleic acid sequence-based amplification (QT-NASBA), strand displacement amplification (SDA), recombinase polymerase amplification (RPA), strand invasion based amplification (SIBA), multiple displacement amplification (MDA), and more recently, cell-free DNA detection [22, 23]. They have substantially increased the sensitivity of detection and quantitation of intestinal nematode infections, especially in low infection intensity settings. They are also highly specific as they can help differentiate between closely related, morphologically identical, human and zoonotic species compared to microscopic techniques which could easily misidentify these parasites [23].

However, various diagnostic methods differ in terms of relevance, cost-effectiveness, ease of operation, sensitivity and specificity [13, 24, 25]. Microscopy has been the most extensively used tool for detecting, recognising, and quantifying parasites for many decades [26]. Currently, the WHO-recommended "gold standard" for the identification of STH eggs is the Kato-Katz method [27]. While parasite microscopy is specific, its sensitivity is dependent on the infection intensity and the time of specimen collection. As nematode eggs can be difficult to detect and identify in low-intensity infections [28], concentration methods are widely used in diagnosis. As nematode eggs have many morphological similarities, microscopy-based detection remains difficult. To be successfully controlled, NTDs must be targeted in an integrated approach [29]. Effective diagnosis should be precise, simple, inexpensive, and the procedures must provide results in a timely manner. Such methods are important in treatment and control programmes, during breakpoints in transmission, and in surveillance for reemergence when possible 'elimination' targets have been achieved [26]. In recent years, there has been a growing effort in some high-income countries to use molecular techniques for STH infections in epidemiological studies and individual patient diagnosis [30, 31]. The main challenge in assessing the accuracy of molecular tests stems from the lack of an optimal gold standard. Even though stool microscopic techniques are commonly used as reference standards, they have relatively low sensitivity. The goal of this review is to generate new statistical evidence for healthcare professionals, researchers, and control strategy developers to use in diagnosing human intestinal nematode infections. Ultimately, this will facilitate improved screening, surveillance and the control and management of human intestinal nematode infections.

## Objective

To determine the diagnostic accuracy of currently available NAATs for human intestinal nematode infections using Kato-Katz as the reference test for the most common STH infections and Scotch tape test for enterobiasis and Baermann method for strongyloidiasis, in different health care settings.

## Methods

As described in this protocol, the methodological approach to evidence searching and synthesis will follow Joanna Briggs Institute (JBI) guidelines on systematic reviews of diagnostic test accuracy [32]. In reporting the findings of this review, standards of the Preferred Reporting Items for Systematic Reviews and Meta-Analyses (PRISMA) will be adhered to [33]. The content of this protocol follows the Preferred Reporting Items for Systematic Review and Meta-Analysis Protocols (PRISMA-P) recommendations [34].

In this review, an initial literature search will be followed by the screening and selection of relevant studies against the preset eligibility criteria. Data extraction, methodological quality assessment, data analyses, evidence generation and report compilation will follow in chronological order.

### Eligibility criteria

**Inclusion criteria.** *Diagnosis of interest.* Human intestinal nematode infections (ascariasis, trichuriasis, hookworm infection, strongyloidiasis and enterobiasis).

*Population.* We will include studies from the community, residential settings, primary, secondary and tertiary care settings. Studies reporting on mixed populations, different geographical regions, and different socioeconomic and social backgrounds will also be included. We will include all asymptomatic and symptomatic people without any restriction on the disease severity. There will be no limitations for gender, age group, ethnicity or country of origin.

*The index tests.* NAATs: Conventional single PCR, nested PCR, real-time PCR, multiplex PCR, and isothermal amplifications assays including LAMP, NASBA, SDA, RPA, SIBA and MDA.

*The reference test.* As no gold standard test is available, we will use WHO recommendations for the review, using the Kato-Katz as the reference test [35–37] for *A. lumbricoides*, *T. trichiura* and hookworm infections. We will use Graham's Scotch tape test for *E. vermicularis* infection, and the Baermann technique for *S. stercoralis* infection as the reference test.

*Outcomes.* The specificity and sensitivity of the index tests are the major outcomes. The probability that the index test will be positive in an infected person is referred to as sensitivity [38]. The probability that the index test will be negative in a non-infected person is known as specificity [38].

**Exclusion criteria.** Case series, commentaries and expert opinions.

Studies that have not investigated the diagnostic accuracy of the index tests.

Studies that have been undertaken on animals.

### Search strategy

An initial limited search of the PubMed online database on the diagnostic test accuracy of available microscopic and molecular methods of human intestinal nematode infections will be carried out. Following the initial search, a review of the keywords found in the titles and abstracts of the extracted feature papers, as well as the key phrases describing the articles, will be performed. Then we will conduct another search using all recognised keywords and index

terms within all included databases, which are PubMed, Google Scholar, CINAHL, Scopus, Trip, Web of Science, and Cochrane Library. Finally, additional articles will be sought from the reference list of retrieved studies. The search terms for PubMed is (((((((((((((("ascaris"[All Fields]) OR ("roundworm"[All Fields])) OR ("necator"[All Fields])) OR ("ancylostoma"[All Fields])) OR ("hookworm"[All Fields])) OR ("strongyloides"[All Fields])) OR ("threadworm"[All Fields])) OR ("trichuris"[All Fields])) OR ("whipworm"[All Fields])) OR ("enterobius"[All Fields])) OR ("pinworm"[All Fields])) OR ("soil transmitted helminth"[All Fields])) OR ("geohelminth"[All Fields])) OR ("intestinal nematode"[All Fields])) AND ((((((((((((((((((((diagnos*) OR ("diagnosis"[All Fields])) OR ("detect"[All Fields])) OR ("screen"[All Fields])) OR (investigat*)) OR ("investigation"[All Fields])) OR ("polymerase chain reaction"[All Fields])) OR ("pcr"[All Fields])) OR ("molecular"[All Fields])) OR ("nucleic acid amplification"[All Fields])) OR ("naat"[All Fields])) OR ("isothermal amplification"[All Fields])) OR ("loop mediated isothermal amplification"[All Fields])) OR ("lamp"[All Fields])) OR ("microscopy"[All Fields])) OR (microscop*)) OR ("kato katz"[All Fields])) OR ("baermann technique"[All Fields])) OR ("scotch tape"[All Fields])) Filters: in the last 10 years, Humans. This approach was developed from earlier pilot searches with the identification of keywords from relevant articles and in consultation with experienced researchers. We will be reviewing the studies undertaken within the last ten years with no language restrictions. Duplicate records will be removed initially with the Mendeley reference management software [39] and then with Rayyan software [40]. The whole text of the identified articles will be examined, with the eligibility criteria to generate the list of final articles.

## Study selection

Findings from the electronic database search will be transferred to Rayyan software [40] for screening and shortlisting based on the inclusion criteria, where any duplicate studies will be removed. The authors will first look at the keywords included in the title and the abstract. Full-text sources of evidence that are relevant will be examined in greater detail based on the inclusion criteria. Two reviewers will work independently to select sources. If there are uncertainties, a third reviewer will be consulted. We will document the rationale for excluding any full-text source information and report it in the systematic review. If a full literature search reveals novel diagnostics that meet the eligibility criteria, these will also be included in the review. This procedure will be recorded, and the methodology will be illustrated in a PRISMA flowchart.

## Assessment of methodological quality

Two reviewers will independently evaluate the risk of bias in each included article and report it according to the Quality Assessment of Diagnostic Accuracy Studies (QUADAS-2) tool [41]. Discrepancies that occur during the process will be resolved by the opinion of a third reviewer. QUADAS-2 includes the risk of bias assessments over four key areas: the patient selection, the index test, the reference standard, and assessment flow and timing.

We will employ the Grading of Recommendations Assessment, Development and Evaluation (GRADE) approach to evaluate the quality of the body of evidence [42]. Accordingly, the body of evidence will be graded as high, moderate, low, or very low quality if data is available [43]. Two reviewers will separately evaluate the body of evidence for each gradable outcome in accordance with the study limitations, consistency of results, directness of evidence, precision, and study design in order to arrive at a grading. Finally, we will put together a summary of the evidence table using the GRADE development tool.

## Data extraction

We will develop a data extraction form using the JBI data extraction instrument as guidance, with modifications relevant to the review. Two reviewers will retrieve information individually, using the customised data extraction form. To ensure consistency, the data extraction protocol will be tested on the first five to ten articles. During data extraction, the customised data extraction format will be amended and revised as needed. Supplementary data from included studies may be retrieved as decided by the review team during the process following the JBI methodology [39]. The revisions will be described in detail in the systematic review report.

Data to be extracted falls into the following domains:

Study identification details: Authors, the year of publication, the country and the funding source.

Study methodological details: Sample size, study design, clinical setting, number of dropouts with reason, diagnostic methods assessed, parasite species studied, assessment of co-infection and any other relevant details like interventions carried out if any.

Population characteristics: Socio-demographic variables (ethnicity, sex, age, religion, marital status, education, employment, income, migration history, household details), history of deworming, and the intensity of infection.

Index test (NAATs) characteristics: The type of test, test design, target selection, the procedure of sample preparation and DNA extraction, output variables produced, quality control measures that were applied during the procedure, sample storage and transport method, the time duration between processing and analysis, and the cost of the test. Depending on the performance of the NAAT analysis, we will rank them accordingly and give different weights during the overall analysis. To give different weights, we will use the GRADE classification [42] as detailed in the methodological quality assessment.

The reference test: As there is no universally accepted "gold" standard, we will use the Kato-Katz as the reference test for the most common STHs, as this test is generally used in published literature and also recommended by the WHO as the most appropriate reference for these species. The way the Kato-Katz has been performed in each study, i.e. number of stool samples taken from a subject for the performance of the test, duration between each stool sample collection, number of smears examined from each stool sample, modifications that were done to the standard protocol, details of the modifications, and quality control measures that were applied during the procedure will be extracted from the selected primary studies. Based on the performance of the Kato-Katz smear examination of the primary study, we will rank each study and give them different weights in the overall analysis. As the reference tests, we will employ the Baermann method for strongyloidiasis and Graham's Scotch tape test for enterobiasis. The number of smears that were evaluated and the timeframe that the sample was collected will be extracted for the Scotch tape sampling. In the Baermann technique, details of the procedure and quality control measures applied will be extracted.

Outcome measures: From contingency tables, we will calculate true positives, false positives, true negatives and false negatives leading to sensitivity, specificity, negative predictive value, positive predictive value, positive likelihood ratio and negative likelihood ratio of index tests.

## Statistical analyses and evidence synthesis

Each index test will be compared against a reference test if data are available. For each test, a true positive, true negative, false positive and false negative will be retrieved. The sensitivity, specificity, positive predictive value, negative predictive value, positive likelihood ratio and negative likelihood ratio of each study will be presented with 95% confidence intervals. They will be shown in paired forest plots. From primary studies, we will estimate summary

sensitivity and specificity using the bivariate model [32]. The bivariate model will directly analyse the association between sensitivity and specificity and consider the random effects to be due to study heterogeneity. According to the availability of data, we will compare index tests using the above-mentioned approach [32]. When heterogeneity is present, it will be visually evaluated by comparing the study results on the paired forest plot. Meta-regression analysis will be used to explore the heterogeneity if five or more studies are available. Review manager software (RevMan web and RevMan 5) [44] will be used to perform this meta-analysis.

## Discussion

Human intestinal nematode infections give rise to a significant global disease burden and high morbidity. Specific and sensitive diagnostic methods are critical in accurately determining worldwide and community burdens and precisely evaluating infection intensities in order to direct effective treatment and other control measures and to track the progress of ongoing STH control programs. Unfortunately, STH infections continue to be underdiagnosed, as they generally tend to be endemic in resource-poor, marginalised communities where there is a lack of trained laboratory staff and limited access to relevant diagnostic procedures. This is further exacerbated by the intermittent shedding of eggs and larvae by STHs which can impact the sensitivity of tests thereby preventing accurate diagnosis. Despite considerable progress in developing effective diagnostic methods, we are still a long way from identifying a gold standard test. Copromicroscopy has been the most extensively used tool for parasite detection and quantification for decades. However, there is a pressing need to avoid repetitive faecal sampling and to employ diagnostic methods with increased levels of sensitivity to assist in the monitoring of control programs, particularly in regions where elimination is the goal. Highly sensitive PCR-based methods can aid in the accurate measurement of STH burdens as control programs are advanced and are extremely useful in detecting worm infections, both in low-endemic areas and in travellers returning from endemic areas. These molecular methods can also diagnose the emergence of new STH variants as they arise. The approaches and methods that will be utilised in guiding this systematic review are outlined in this protocol. This is an important phase in the review process because it ensures that the final review has been well-planned and recorded to ensure accountability and transparency. It will also allow future researchers to replicate the review procedures [34]. The primary goal of this systematic review is to evaluate NAATs available for detecting human intestinal nematodes, including their sensitivity and specificity, and their strengths and limitations. This review will be focused on evidence-based conclusions and will provide key up-to-date information for researchers, clinicians and policymakers on the performance of microscopic and molecular-based approaches for the diagnosis of human intestinal nematode infections.

## Supporting information

**S1 Checklist. PRISMA–P 2015 checklist.** Preferred reporting items for systematic review and meta-analysis protocols 2015 checklist: Recommended items to address in a systematic review protocol.
(PDF)

## Author Contributions

**Conceptualization:** Nalini Kaushalya Jayakody, Kosala Gayan Weerakoon.

**Methodology:** Nalini Kaushalya Jayakody, Poornima Lakshini Kumbukgahadeniya, Anjana Silva, Nuwan Darshana Wickramasinghe, Susiji Wickramasinghe, Donald Peter McManus, Kosala Gayan Weerakoon.

**Project administration:** Kosala Gayan Weerakoon.

**Supervision:** Anjana Silva, Nuwan Darshana Wickramasinghe, Susiji Wickramasinghe, Donald Peter McManus, Kosala Gayan Weerakoon.

**Writing – original draft:** Nalini Kaushalya Jayakody, Kosala Gayan Weerakoon.

**Writing – review & editing:** Nalini Kaushalya Jayakody, Poornima Lakshini Kumbukgahadeniya, Anjana Silva, Nuwan Darshana Wickramasinghe, Susiji Wickramasinghe, Donald Peter McManus, Kosala Gayan Weerakoon.

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
