## [Decision Letter · Decision Letter 0]

12 Aug 2022

PONE-D-22-16144

The accuracy of PCR-based methods in predicting human intestinal nematode infections: a protocol for a systematic review and meta-analysis of the accuracy of available diagnostic tests

PLOS ONE

Dear Dr. Weerakoon,

Thank you for submitting your manuscript to PLOS ONE. After careful consideration, we feel that it has merit but does not fully meet PLOS ONE’s publication criteria as it currently stands. Therefore, we invite you to submit a revised version of the manuscript that addresses the points raised during the review process.

Please find the comments provided by the reviewers below this email.

We look forward to receiving your revised manuscript.

Kind regards,

Hesham

Hesham M. Al-Mekhlafi, PhD

Academic Editor

PLOS ONE

Journal Requirements:

 "This review is a part of an intestinal parasitoses community evaluation project funded by the National Research Council of Sri Lanka (NRC 20-118). However, this particular publication (review protocol) is not funded through this grant. The funder had and will not have a role in study design, data collection and analysis, decision to publish, or preparation of the manuscript."

Reviewers' comments:

Reviewer's Responses to Questions

**Comments to the Author**

1. Does the manuscript provide a valid rationale for the proposed study, with clearly identified and justified research questions?

Reviewer #1: Partly

Reviewer #2: Yes

2. Is the protocol technically sound and planned in a manner that will lead to a meaningful outcome and allow testing the stated hypotheses?

Reviewer #1: Partly

Reviewer #2: Yes

3. Is the methodology feasible and described in sufficient detail to allow the work to be replicable?

Reviewer #1: No

Reviewer #2: Yes

4. Have the authors described where all data underlying the findings will be made available when the study is complete?

Reviewer #1: Yes

Reviewer #2: Yes

5. Is the manuscript presented in an intelligible fashion and written in standard English?

Reviewer #1: Yes

Reviewer #2: Yes

6. Review Comments to the Author

You may also provide optional suggestions and comments to authors that they might find helpful in planning their study.

Reviewer #1: This manuscript describes a protocol to perform a systematic review, analysing the diagnostic accuracy of PCR-based methods for the detection of human intestinal nematodes. Taking into account the increasing use of nucleic acid amplification tests (NAAT’s) in the diagnosis of intestinal helminths, I think the authors have chosen a timely topic for their review. However, I see several flaws and inconsistencies in their approach.

First of all, what I understand from the methods section is that the Kato-Katz smear will be used as the reference test. Therefore, clarity should be given about the way the Kato-Katz has been performed in each study, i.e. how many smears have been examined, based on how many stool samples and were there appropriate quality control measures in place. Based on the described performance of the Kato-Katz smear examination the authors could rank each individual study included and give them a different weight in the overall analysis.

Furthermore, for Strongyloides and Enterobius, it is not clear to me how the authors want to compare the PCR outcome with Kato-Katz data. For these two species other microscopy procedures (Baermann, tape-test) are more appropriate.

In addition the authors should be more specific about the NAAT’s included. Even if limiting to a real-time PCR format, each test is different, be it either in design and target selection, as well as in procedures of sample preparation and DNA extraction, the output variable producing, and most importantly, the level of appropriate quality control measures implemented. How to interpret these differences between publications is not clarified in this protocol. The description of Line 187-188 I find therefore far from complete.

This lack of uniformity is not only a major struggle for the analysis of PCR performance, but also for serology. This brings me to another point. In the abstract I read: “We plan to analyse the efficacy of microscopic- and PCR-based approaches in detecting human intestinal nematode infections in different health care settings.” In the key-words I noticed “serology”, but nothing is mentioned in the abstract. Only from the search strategy (line 225) and description at line 181-186 I understand that several immune-diagnostic tests will be included as index test, including antibody tests and antigen tests. This sounds rather ambitious to me. The authors should be more clear what the relevance is of including immunodiagnostic test and how they want to summarise the different tests available. My advise is to leave the immunodiagnostic tests out of the analysis.

Also for the different microscopy-based procedures included, the descriptions are not very clear to me. In my opinion the protocol would improve if the focus would be on a comparison between real-time PCR as the index test and the most optimal microscopy procedure as a reference test. Concerning microscopy, this is Kato-Katz for most STH species and species-dedicated procedures for Strongyloides and Enterobius

Minor comments:

Abstract: the proposed reference test is not described (as it is in line 138-140).

Line 62: other Ancylostoma species should also be included .

Line 226: ELIZA should be ELISA.

Line 226: “immunofluorescent antibody” is not an appropriate search term.

Line 278: “The gold standard: As there is no universally accepted gold standard, we will use those in the published literature and the WHO recommended gold standard, which is Kato Katz.” I suggest to rephrase this as follows: “Reference test: As there is no universally accepted “gold” standard, we will use the Kato Katz as the reference test for the most common soil transmitted helminths, as this test is generally used in published literature and also recommended by the WHO as the most appropriate reference for these species.”

Reviewer #2: Title: The accuracy of PCR-based methods in predicting human intestinal nematode infections: a protocol for a systematic review and meta-analysis of the accuracy of available diagnostic tests.

Comments

This is a well thought out protocol, focusing on diagnostic techniques. It is well written with clear and concise methods which are adequate to realise the study goals. However, I wish to raise a few issues that the authors should consider before publication of the protocol.

1. From the study objective, and the narrative in the whole protocol, it is clear that the goal of the systematic review is to evaluate different diagnostic techniques/ approaches that are available for evaluation of human intestinal nematodes. However, the title of the study is more biased to PCR-based approaches which are not the primary focus of the study. I will suggest that the title is reviewed to capture the whole study and not part of the study.

2. In the objective (Abstract section), the authors states that the study will specifically focus on microscopic and PCR based approaches, however in the Eligibility criteria (index tests Page 7 and 8), they propose to include serology based assays. This is a key area in diagnosis of human nematodes and needs to come out well in the objectives and also in the title.

7. PLOS authors have the option to publish the peer review history of their article (what does this mean?). If published, this will include your full peer review and any attached files.

Reviewer #1: No

Reviewer #2: No

---

## [Author Response · Author response to Decision Letter 0]

23 Sep 2022

Journal Requirements:

and https://journals.plos.org/plosone/s/file?id=ba62/PLOSOne_formatting_sample_title_authorsaffiliations.pdf

Our response – Revised as requested adhering to the style guidelines. 

“This review is a part of an intestinal parasitoses community evaluation project funded by the National Research Council of Sri Lanka (NRC 20-118). However, this particular publication (review protocol) is not funded through this grant. The funder had and will not have a role in study design, data collection and analysis, decision to publish, or preparation of the manuscript."

Our response - We have included the relevant details of the financial disclosure in the cover letter.

Reviewers' comments:

Our response to reviewer # 1 

We thank the reviewer for his kind comments and useful insights. We found them very helpful to improve our protocol for the systematic review.

Reviewer #1: This manuscript describes a protocol to perform a systematic review, analysing the diagnostic accuracy of PCR-based methods for the detection of human intestinal nematodes. Taking into account the increasing use of nucleic acid amplification tests (NAAT’s) in the diagnosis of intestinal helminths, I think the authors have chosen a timely topic for their review. However, I see several flaws and inconsistencies in their approach.

1. First of all, what I understand from the methods section is that the Kato-Katz smear will be used as the reference test. Therefore, clarity should be given about the way the Kato-Katz has been performed in each study, i.e. how many smears have been examined, based on how many stool samples and were there appropriate quality control measures in place. Based on the described performance of the Kato-Katz smear examination the authors could rank each individual study included and give them a different weight in the overall analysis.

Our response - As suggested, we have updated the data extraction phase of the Reference test in the methods section, to include information on the performance of the Kato-Katz, i.e.

number of stool samples taken from a subject for the performance of the test, the duration between each stool sample collection, number of smears examined from each stool sample, whether any modifications were done to the standard protocol, and the important details of the modifications done, quality control measures applied during the Kato-Katz procedure. Accordingly, when we are carrying out the data extraction of the selected studies, we will extract all of the above-mentioned details from the primary study. Based on the performance of the Kato-Katz smear examination of the primary study, we will rank each study and give them different weights in the overall analysis. (Line 241-247)

2. Furthermore, for Strongyloides and Enterobius, it is not clear to me how the authors want to compare the PCR outcome with Kato-Katz data. For these two species other microscopy procedures (Baermann, tape-test) are more appropriate.

Our response - We have added the Baermann test and the Graham Scotch tape tests as reference tests for strongyloidiasis and enterobiasis, respectively. (Line 159,160,247-252)

3. In addition the authors should be more specific about the NAAT’s included. Even if limiting to a real-time PCR format, each test is different, be it either in design and target selection, as well as in procedures of sample preparation and DNA extraction, the output variable producing, and most importantly, the level of appropriate quality control measures implemented. How to interpret these differences between publications is not clarified in this protocol. The description of Line 187-188 I find therefore far from complete. This lack of uniformity is not only a major struggle for the analysis of PCR performance, but also for serology.

Our response - As proposed, we have revised the Index test section of the data extraction to include the information on the NAATs, i.e test design, target selection, the procedure of sample preparation and DNA extraction, output variables produced, whether appropriate quality control measures were applied during the procedure Depending on the performance of the NAATs analysis, we will rank them accordingly and give different weights during the overall analysis. ( Line 232-237)

4. This brings me to another point. In the abstract I read: “We plan to analyse the efficacy of microscopic- and PCR-based approaches in detecting human intestinal nematode infections in different health care settings.” In the key-words I noticed “serology”, but nothing is mentioned in the abstract. Only from the search strategy (line 225) and description at line 181-186 I understand that several immune-diagnostic tests will be included as index test, including antibody tests and antigen tests. This sounds rather ambitious to me. The authors should be more clear what the relevance is of including immunodiagnostic test and how they want to summarise the different tests available. My advise is to leave the immunodiagnostic tests out of the analysis.

Our response - As suggested, we have dropped the analysis of immunodiagnostic tests from the protocol. As a result, the exclusion of the serology from the abstract and objectives is now addressed. Accordingly, we have removed “serology” from the keyword list and the search string.

5. Also for the different microscopy-based procedures included, the descriptions are not very clear to me. In my opinion the protocol would improve if the focus would be on a comparison between real-time PCR as the index test and the most optimal microscopy procedure as a reference test. Concerning microscopy, this is Kato-Katz for most STH species and species-dedicated procedures for Strongyloides and Enterobius

Our response - As suggested, we have removed all the other microscopic diagnostic methods as index tests.

6. Abstract: the proposed reference test is not described (as it is in line 138-140).

Our response - The reference tests were described in the abstract as suggested. (line 34-37)

7. Line 62: other Ancylostoma species should also be included.

Our response – Revised as suggested (revised line 54)

8. Line 226: ELIZA should be ELISA. 

Our response - Serology was removed from the protocol. Accordingly, ELISA was also removed from this line. 

9. Line 226: “immunofluorescent antibody” is not an appropriate search term.

Our response - As serology was removed from the protocol, the immunofluorescent antibody was also removed from this line.

10. Line 278: “The gold standard: As there is no universally accepted gold standard, we will use those in the published literature and the WHO recommended gold standard, which is Kato- Katz.” I suggest to rephrase this as follows: “Reference test: As there is no universally accepted “gold” standard, we will use the Kato -Katz as the reference test for the most common soil transmitted helminths, as this test is generally used in published literature and also recommended by the WHO as the most appropriate reference for these species.”

Our response - This line was revised as suggested. ( revised line 238- 241)

Reviewer #2: Title: The accuracy of PCR-based methods in predicting human intestinal nematode infections: a protocol for a systematic review and meta-analysis of the accuracy of available diagnostic tests.

Comments

This is a well thought out protocol, focusing on diagnostic techniques. It is well written with clear and concise methods which are adequate to realise the study goals. However, I wish to raise a few issues that the authors should consider before publication of the protocol.

Our response to reviewer # 2

We thank the reviewer for appreciating our work and value the comments and suggestions given by the reviewer. Our responses to the comments are stated below.

1. From the study objective, and the narrative in the whole protocol, it is clear that the goal of the systematic review is to evaluate different diagnostic techniques/ approaches that are available for evaluation of human intestinal nematodes. However, the title of the study is more biased to PCR-based approaches which are not the primary focus of the study. I will suggest that the title is reviewed to capture the whole study and not part of the study.

Our response - As the first reviewer has suggested, we have taken serology out of our protocol as an index test. So we did not include any serology-related details in the abstract. Therefore, we feel that there is no necessity of amending the title.

2. In the objective (Abstract section), the authors states that the study will specifically focus on microscopic and PCR based approaches, however in the Eligibility criteria (index tests Page 7 and 8), they propose to include serology based assays. This is a key area in diagnosis of human nematodes and needs to come out well in the objectives and also in the title.

Our response - As stated above, we have taken serology out of our protocol as an index test. Thus, we did not include any serology-related information in the abstract and the objectives.

---

## [Decision Letter · Decision Letter 1]

12 Oct 2022

PONE-D-22-16144R1The accuracy of nucleic acid amplification tests (NAATs) in detecting human intestinal nematode infections: a protocol for a systematic review and meta-analysisPLOS ONE

Dear Dr. Weerakoon,

Thank you for submitting your manuscript to PLOS ONE. After careful consideration, we feel that it has merit but does not fully meet PLOS ONE’s publication criteria as it currently stands. Therefore, we invite you to submit a revised version of the manuscript that addresses the points raised during the review process.

Please find few minor comments by the reviewers at the end of this email. Moreover, please prepare the list of references according to the journal's instruction and style.==============================

We look forward to receiving your revised manuscript.

Kind regards,

Hesham

Hesham M. Al-Mekhlafi, PhD

Academic Editor

PLOS ONE

Journal Requirements:

Reviewers' comments:

Reviewer's Responses to Questions

**Comments to the Author**

1. Does the manuscript provide a valid rationale for the proposed study, with clearly identified and justified research questions?

Reviewer #1: Yes

2. Is the protocol technically sound and planned in a manner that will lead to a meaningful outcome and allow testing the stated hypotheses?

Reviewer #1: Yes

3. Is the methodology feasible and described in sufficient detail to allow the work to be replicable?

Reviewer #1: Yes

4. Have the authors described where all data underlying the findings will be made available when the study is complete?

Reviewer #1: Yes

5. Is the manuscript presented in an intelligible fashion and written in standard English?

Reviewer #1: Yes

6. Review Comments to the Author

You may also provide optional suggestions and comments to authors that they might find helpful in planning their study.

Reviewer #1: I read the rebuttal and reviewed the changed made in the manuscript. In my opinion the authors responded well to all points mentioned by the reviewers. In particular I appreciate the more focused approach of the protocol.

Still, I have a number of technical comments. Most of these comments relate to the NAAT-procedures. The authors could be more precise in their wording and definitions at several points throughout the manuscript.

The line numbers mentioned refer to the manuscript including track changes.

Abstract (line 40): “… to more specialised high sequencing technologies such as the NAAT.” It is not clear to me what the authors try to say here. NAAT is a very broad definition of all tests related to nucleic acid amplification. Consequently “the” (?) NAAT is not an example of sequencing. More the other way around. Furthermore, the abbreviation NAAT is explained in the title. Still this should be repeated in the abstract (and in the introduction, see below).

Abstract (line 53): indeed, the details of the data bases are not so relevant for an abstract.

Objectives (line 151-152): I agree that the objectives improved by using NAATs and not “PCR-based methods”. But at the introduction section the terminology of NAATs have not been well introduced. In line 121-122 PCR is introduced as a method, but nothing is mentioned about other NAAT-methods, including isothermal amplification procedures, such as LAMP.

Inclusion criteria (line 187-188): LAMP is one example of several isothermal amplification procedures. It is not clear to me why only LAMP is mentioned an no other methods.

Search strategy (line 237-238): again it is not clear to me why only “loop mediated isothermal amplification” OR “LAMP” is included in the search terms for PubMed. I suggest to include the term “isothermal amplification” as well.

Data extraction (line 326-327): I appreciate the inclusion of a weight-factor, based on the quality of the used diagnostic procedures. But the description how this is done is completely lacking.

Data extraction (line 346-347): ”…the number of smears taken”. Please explain. I have never heard of “smears” in the context of a Baermann procedure. In addition ‘the standard technique” is not existing for the Baermann. Rather different approaches have been described.

7. PLOS authors have the option to publish the peer review history of their article (what does this mean?). If published, this will include your full peer review and any attached files.

Reviewer #1: No

---

## [Author Response · Author response to Decision Letter 1]

22 Nov 2022

Response to reviewers: Title: The accuracy of nucleic acid amplification tests (NAATs) in detecting human intestinal nematode infections: a protocol for a systematic review and meta-analysis

To,

The Editor/Reviewers,

PLOS ONE

Date: November 22, 2022

Subject: Addressing Editor’s/Reviewers’ Comments for PONE-D-22-16144

We appreciate the editor's and the reviewers' constructive suggestions on our revised protocol. We have attempted to address all the issues brought up by the academic editor and the reviewers. We believe that all the issues are satisfactorily addressed.

Sincerely on behalf of all the authors

Prof Kosala Weerakoon

Journal Requirements:

Response – The modifications we made to the reference section are listed below.

We have added new references numbering 21, 22, 23, 42 and 43 and accordingly references 39 and 41 of the previous list were removed to avoid repetition. We have not added any retraced papers.

Our response to reviewer # 1 

We appreciate the reviewer's thoughtful remarks and insightful observations. They were incredibly useful to us in enhancing our systematic review protocol.

The line numbers we've provided correspond to those in the clean manuscript.

Reviewer #1: I read the rebuttal and reviewed the changes made in the manuscript. In my opinion the authors responded well to all points mentioned by the reviewers. In particular I appreciate the more focused approach of the protocol.

Still, I have a number of technical comments. Most of these comments relate to the NAAT-procedures. The authors could be more precise in their wording and definitions at several points throughout the manuscript.

The line numbers mentioned refer to the manuscript including track changes.

1. Abstract (line 40): “… to more specialised high sequencing technologies such as the NAAT.” It is not clear to me what the authors try to say here. NAAT is a very broad definition of all tests related to nucleic acid amplification. Consequently “the” (?) NAAT is not an example of sequencing. More the other way around. Furthermore, the abbreviation NAAT is explained in the title. Still this should be repeated in the abstract (and in the introduction, see below).

Response – We have explained the NAATs in the abstract and also in the introduction section (lines 32-36, 109-121)

Abstract (lines32-36)

“Molecular methods use NAA to amplify the deoxyribonucleic acid (DNA) or ribonucleic acid (RNA) fragments of the parasite to detect and determine its presence using different technologies (NAAT). They have increased the sensitivity of detection and quantitation of intestinal nematode infections, especially in low infection intensity settings.”

Introduction (lines 109 -121)

“NAATs are different molecular methods designed to amplify specific DNA or RNA fragments into a higher number of copies in order to determine and characterize their presence. Types of NAATs used for the diagnosis of intestinal nematode infections are polymerase chain reaction (PCR), including conventional single, nested, real-time, multiplex PCR and digital PCR, isothermal assays like loop-mediated isothermal amplification (LAMP) quantitative nucleic acid sequence-based amplification (QT-NASBA), strand displacement amplification (SDA), Recombinase polymerase amplification (RPA), Strand invasion based amplification (SIBA), Multiple displacement amplification (MDA), and more recently, cell-free DNA detection. They have substantially increased the sensitivity of detection and quantitation of intestinal nematode infections, especially in low infection intensity settings. They are also highly specific as they can help differentiate between closely related, morphologically identical, human and zoonotic species compared to microscopic techniques which could easily misidentify these parasites”

2. Abstract (line 53): indeed, the details of the data bases are not so relevant for an abstract.

 Response – As suggested, we have removed the details of the databases from the abstract

3. Objectives (lines 151-152): I agree that the objectives improved by using NAATs and not “PCR-based methods”. But at the introduction section the terminology of NAATs have not been well introduced. In line 121-122 PCR is introduced as a method, but nothing is mentioned about other NAAT-methods, including isothermal amplification procedures, such as LAMP.

Response – We have introduced the word NAAT and provided details of the NAATs carried out in the diagnosis of human intestinal nematode infections. (lines 109-121)

Introduction (lines 109 -121)

“NAATs are different molecular methods designed to amplify specific DNA or RNA fragments into a higher number of copies in order to determine and characterize their presence. Types of NAATs used for the diagnosis of intestinal nematode infections are polymerase chain reaction (PCR), including conventional single, nested, real-time, multiplex PCR and digital PCR, isothermal assays like loop-mediated isothermal amplification (LAMP) quantitative nucleic acid sequence-based amplification (QT-NASBA), strand displacement amplification (SDA), Recombinase polymerase amplification (RPA), Strand invasion based amplification (SIBA), Multiple displacement amplification (MDA), and more recently, cell-free DNA detection. They have substantially increased the sensitivity of detection and quantitation of intestinal nematode infections, especially in low infection intensity settings. They are also highly specific as they can help differentiate between closely related, morphologically identical, human and zoonotic species compared to microscopic techniques which could easily misidentify these parasites”

4. Inclusion criteria (lines 187-188): LAMP is one example of several isothermal amplification procedures. It is not clear to me why only LAMP is mentioned and no other methods.

Response – As proposed, we have included other isothermal amplification methods to the inclusion criteria other than LAMP (lines172 and 173)

Inclusion criteria (lines 172 and 173)

“NAATs: Conventional single PCR, nested PCR, real-time PCR, multiplex PCR, and isothermal amplifications assays including LAMP, NASBA, SDA, RPA, SIBA and MDA”

5. Search strategy (lines 237-238): again it is not clear to me why only “loop-mediated isothermal amplification” OR “LAMP” is included in the search terms for PubMed. I suggest to include the term “isothermal amplification” as well.

Response – As suggested, we have included “isothermal amplification” to the search string (lines 206)

6. Data extraction (lines 326-327): I appreciate the inclusion of a weight-factor, based on the quality of the used diagnostic procedures. But the description of how this is done is completely lacking.

Response – We will use GRADE classification to give different weights to the studies based on the quality of the diagnostic procedure. In the methodological quality assessment section, we have introduced the term GRADE and given the details of the classification and how it will be utilized to give different weights to the studies based on the quality of the diagnostic procedure employed. (lines 233-239)

 (Lines 233-239)

“We will employ the Grading of Recommendations Assessment, Development and Evaluation (GRADE) approach to evaluate the quality of the body of evidence. Accordingly, the body of evidence will be graded as high, moderate, low, or very low quality if data is available. Two reviewers will separately evaluate the body of evidence for each gradable outcome in accordance with the study limitations, consistency of results, directness of evidence, precision, and study design in order to arrive at a grading. Finally, we will put together a summary of the evidence table using the GRADE development tool. 

6. Data extraction (line 346-347): ”…the number of smears taken”. Please explain. I have never heard of “smears” in the context of a Baermann procedure. In addition ‘the standard technique” is not existing for the Baermann. Rather different approaches have been described.

Response – We have removed “the number of smears taken” and “the standard technique” in the data extraction section of the Baermann test and included the “details of the different approaches of the Baermann test”.(lines 277 and 278)

---

## [Editor Report · Decision Letter 2]

24 Nov 2022

The accuracy of nucleic acid amplification tests (NAATs) in detecting human intestinal nematode infections: a protocol for a systematic review and meta-analysis

PONE-D-22-16144R2

Dear Dr. Weerakoon,

We’re pleased to inform you that your manuscript has been judged scientifically suitable for publication and will be formally accepted for publication once it meets all outstanding technical requirements.

Kind regards,

Hesham

Hesham M. Al-Mekhlafi, PhD

Academic Editor

PLOS ONE

Additional Editor Comments (optional):

This version has addressed all comments highlighted by the reviewers.

I would like to take this opportunity to express my deepest sympathy and condolences on the loss of Prof. Donald McManus of QIMR Berghofer Research Institute, Australia, to his family, friends, students, colleagues and the science community. Prof. Donald McManus was an esteemed parasitologist who dedicated his life to the prevention and elimination of neglected tropical diseases, and has made incredible contributions to the field of parasitology. Rest in paece!

Hesham Al-Mekhlafi
---

## [Editor Report · Acceptance letter]

2 Dec 2022

PONE-D-22-16144R2 

The accuracy of nucleic acid amplification tests (NAATs) in detecting human intestinal nematode infections: a protocol for a systematic review and meta-analysis 

Dear Dr. Weerakoon:

I'm pleased to inform you that your manuscript has been deemed suitable for publication in PLOS ONE. Congratulations! Your manuscript is now with our production department. 

Kind regards, 

on behalf of

Prof. Hesham M. Al-Mekhlafi 

Academic Editor

PLOS ONE